# Urinary Fluoride Levels Among Youth in the National Health and Nutrition Examination Survey (NHANES) 2015–2016: Potential Differences According to Race

**DOI:** 10.3390/nu17020309

**Published:** 2025-01-16

**Authors:** Durdana Khan, Stephen Franks, Zhilin Wang, Angela Miles, Howard Hu, Ashley J. Malin

**Affiliations:** 1College of Public Health and Health Professions, University of Florida, Gainesville, FL 32611, USA; khan.durdana@ufl.edu (D.K.); stephenfranks@phhp.ufl.edu (S.F.); z.wang6@ufl.edu (Z.W.); a.miles@phhp.ufl.edu (A.M.); 2College of Medicine, University of Florida, Gainesville, FL 32610, USA; 3Department of Population and Public Health Sciences, Keck School of Medicine of University of Southern California, Los Angeles, CA 90032, USA; howard.hu@med.usc.edu

**Keywords:** fluoride, urine fluoride, tap water, United States, children, adolescents, race/ethnicity, NHANES

## Abstract

Background/Objectives: Urinary fluoride (UF) is the most well-established biomarker for fluoride exposure, and understanding its distribution can inform risk assessment for potential adverse systemic health effects. To our knowledge, this study is the first to report distributions of UF among youth according to sociodemographic factors in a nationally representative United States (US) sample. Methods: The study included 1191 children aged 6-11 years and 1217 adolescents aged 12-19 years from the National Health and Nutrition Examination Survey (NHANES) 2015–2016. We examined UF according to sociodemographic variables as well as Spearman correlations between UF and plasma fluoride. Survey-weighted quantile regression examined associations between tap water fluoride and UF levels adjusted for covariates. Results: The average age of participants was 12.5 years. The median (IQR) UF and water fluoride concentrations were 0.52 (0.50) mg/L and 0.39 (0.54) mg/L, respectively. Children had higher UF levels than adolescents and males had higher UF levels than females. UF differed according to race/ethnicity among both children and adolescents. Specifically, non-Hispanic Black youth tended to have higher UF levels than all participants except for those classified as other race/multiracial. UF and plasma fluoride were moderately correlated for children and adolescents. Higher water fluoride levels were associated with higher UF levels, and the magnitudes of association were larger at higher quantiles of UF (β = 0.14, *p* < 0.001; β = 0.20, *p*< 0.001 at the 25th and 50th quantiles, respectively). The magnitude of association between water fluoride and UF was the largest for non-Hispanic Black participants (predictive margin = 0.3, *p* < 0.001). Conclusions: Non-Hispanic Black youth in the US may have greater fluoride exposure and receive more of their fluoride intake from tap water than youth of other races/ethnicities. Factors contributing to potential racial/ethnic disparities in fluoride exposure within the US warrant further investigation so that they can be mitigated to reduce the potential for harm.

## 1. Introduction

Fluoride is an environmentally ubiquitous mineral [1]. It is added to oral health products, and can also supplement salt, milk or drinking water for the prevention of dental caries [2,3]. In 1945, Grand Rapids, Michigan, became the first United States (US) city to implement community water fluoridation [4]. Since then, community fluoridation has become widespread. Currently, nearly three-quarters of the US population on community drinking water systems is administered fluoridated water [2]. The targeted fluoride concentration for protecting against dental caries, while minimizing risk of dental fluorosis, is 0.7 mg/L [5]. Although fluoride can improve dental health, concerns have been raised that it can also contribute to adverse health effects for bone, as well as endocrine and organ systems, including the brain [6,7]. Increased fluoride exposure at US-population-relevant levels has been shown in a number of epidemiological cohort studies to be associated with adverse neurocognitive developmental outcomes in children [8,9,10,11]. Additionally, recent epidemiological studies have observed that higher water and/or plasma fluoride levels are associated with poorer sleep health, markers of decreased renal clearance, and earlier menarche among US adolescents in the National Health and Nutrition Examination Survey (NHANES) [12,13,14]. Although these studies included plasma fluoride as a biomarker of fluoride exposure, urinary fluoride (UF), a measure of total fluoride intake, is considered the most well-established and widely used fluoride exposure biomarker, particularly for studies of child neurodevelopment [1]. Moreover, a recent meta-analysis conducted by the National Toxicology Program found that higher early life fluoride exposure is inversely associated with child IQ in countries outside of the US; however, they noted that they could not apply these findings to the US population because no nationally representative urinary fluoride levels are available [15]. Thus, understanding distributions of UF according to sociodemographic parameters in a nationally representative sample can inform risk assessment for adverse systemic health effects of fluoride and clarify potentially vulnerable subpopulations.

In November 2022, NHANES released the first nationally representative data on UF levels among children and adolescents in the 2015–2016 cycle. The current study is the first to characterize these UF levels according to sociodemographic factors and in relation to other fluoride exposure measures. We explored associations of UF with sociodemographic factors and hypothesized that plasma and water fluoride levels would each be positively associated with UF rather than negatively or not associated with it.

## 2. Materials and Methods

### 2.1. Participants

The study included children and adolescents from the National Health and Nutrition Examination Survey (NHANES) 2015–2016 cycle. NHANES is a collection of studies conducted by the National Center for Health Statistics (NCHS) assessing health and diet among approximately 5000 children and adults of all ages across the US. Information about health, diet, and sociodemographic variables is collected via interviews, while blood and urine tests are collected along with dental assessments and body measurements at the mobile examination center (MEC) [16]. Informed consent or assent is provided by all study participants. Participants aged 18 years and over, as well as emancipated adolescents aged 16 years and older provide consent for themselves for the interview and MEC visit. Parents/guardians provide consent for minors for the interview. For the MEC visit, parental consent and child assent are required for children aged 7–17 years while parental consent is provided for children 0–6 years of age [17].

The NCHS Ethics Review Board, and the formal review boards that preceded it, have approved each continuous NHANES cycle [18]. The 2015–2016 cycle was the only cycle that had publicly available data on UF levels, and UF data were only provided for youth aged 6–19 years. There were 2408 youth, including 1191 children aged 6–11 years and 1217 adolescents aged 12–19 years who had UF measured and were included in univariate analyses. There were 2356 participants who had both urine and household tap water fluoride measurements. However, we excluded 359 participants who reported not drinking tap water when examining associations of water fluoride and UF. This resulted in a sample of 1997 for these analyses. In addition, for the final study analysis, we excluded 187 participants with missing data for covariates (<10%t) (see Appendix A for a participant selection flow diagram). Therefore, the total number of participants excluded due to missing information for urinary fluoride (*n* = 139), water fluoride (*n* = 52), and covariates (*n* = 187) was 378. There were no appreciable differences in demographic characteristics between the final study sample (N = 1810) and the study sample including all those with missing data (N = 2188) (see Appendix A). This study was exempt from IRB review by the University of Florida (Protocol #: ET00021469).

### 2.2. Measures

#### 2.2.1. Urinary Fluoride (UF)

Fluoride concentrations in urine samples were measured using an ion-selective electrode (ISE) along with a total ionic strength adjustment buffer (TISAB). The TISAB adjusts the ionic strength of fluoride with sodium chloride and buffers the solution to a pH of 5. Urinary fluoride analyses was conducted at the Division of Laboratory Sciences, National Center for Environmental Health, Centers for Disease Control and Prevention, Atlanta, GA. [19]. Samples were excluded from UF measurement if there was suspected contamination during collection, contamination during analysis, or an insufficient volume [19]. The National Center for Health Statistics (NCHS) imputed values below the lower limit of detection (LLOD) of 0.144 mg/L by dividing LLOD by the square root of 2 (LLOD/sqrt [2]). Approximately 6% of UF samples were below the LLOD [19]. We calculated creatinine adjusted UF concentrations (UF_CR_) for comparison with unadjusted UF. UF_CR_ values were calculated separately for children and adolescents using the following formula [9,20]:(1)$$\left[\frac{Fluoride\;concentration\;in\;urine\;sample}{Creatinine\;concentration\;in\;urine\;sample}\right]\times \;\mathrm{The}\;\mathrm{average}\;\mathrm{creatinine}\;\mathrm{concentration}\;\mathrm{of}\;\mathrm{the}\;\mathrm{study}\;\mathrm{sub}\text{-}\mathrm{sample}$$

However, we included unadjusted UF measurements in our primary analyses, as urinary creatinine varies according to sociodemographic factors, including age, race/ethnicity, and BMI, and increases with age across childhood [21]. Furthermore, adjusting UF for urinary dilution using urinary creatinine in a racially heterogeneous sample can introduce bias due to differences in urinary creatinine according to race [22].

#### 2.2.2. Plasma Fluoride and Tap Water Fluoride

Plasma fluoride was measured using an ISE along with a hexamethyldisiloxane-facilitated diffusion method to increase the concentration of fluoride in the solution. This method ensures that all plasma fluoride concentrations analyzed are above the LOD and linearly proportional on the standard curve [23]. The LLOD for plasma fluoride was 0.25 nmol. Fluoride was measured in tap water samples collected in participants’ homes using an ISE after the tap water flowed for 5–10 s [24]. The LLOD for water fluoride was 0.10 mg/L. NCHS imputed values below the LLOD by dividing the LLOD by the square root of 2 (LLOD/sqrt [2]). Approximately 31% of plasma fluoride samples and 12% of water fluoride samples were below the LLOD [23,24]. Fluoride in plasma and water samples was analyzed at the College of Dental Medicine, Georgia Regents University, Augusta, GA. It was measured in duplicate in the same sample and the average fluoride concentration was publicly released. Samples were excluded if there were methodological issues during collection that led to elevated fluoride readings, they had insufficient volume, or were thawed for more than one day [23,24]. Fluoride content in participants’ municipal water or foods and beverages, as well as community fluoridation status, was not publicly reported in NHANES 2015–2016.

#### 2.2.3. Sociodemographic Variables

We included sociodemographic variables that have been associated with UF levels in children and adolescents, as well as with fluoride excretion and metabolism in previous studies [21,25,26]. We considered the race/ethnicity, age, sex, ratio of family income to poverty, and body mass index (BMI).

##### Age

Age in years was determined from the participant’s birthdate provided during the survey interview. For cases in which the date of birth was not available, self-reported age in years was provided. We stratified the study sample by child and adolescent age ranges provided by the CDC (6–11 years for children and 12–19 years for adolescents).

##### Race/Ethnicity

Participant race and ethnicity were ascertained via a questionnaire [27]. Respondents were classified as either “Mexican American”, “Other Hispanic”, “non-Hispanic White”, “non-Hispanic Black”, “non-Hispanic Asian”, and “Other Race, Including non-Hispanic Multiracial” [27]. “Other Race” includes all those who identify themselves as belonging to more than one race.

##### Body Mass Index (BMI)

BMI in kg/m^2^ was ascertained from participant height and weight. It was measured continuously and categorically. The NCHS utilized age- and sex-specific percentiles of 2000 CDC growth charts to identify BMI categories for youth [28]. BMI was categorized as underweight (<5th percentile), normal weight (5th to <85th percentiles), overweight (85th to <95th percentiles), and obese (≥95th percentile).

##### Ratio of Family Income to Poverty

The ratio of family income to poverty was calculated by dividing annual family income by the poverty guidelines for the survey year. The Department of Health and Human Services (HHS) poverty guidelines were used as the poverty measure to calculate this ratio [27]. The values range from 0–5 and were not computed if family income data were missing.

#### 2.2.4. Statistical Analysis

We conducted univariate analysis to examine descriptive statistics for participant demographic characteristics and fluoride variables. Fluoride variables were right skewed. Therefore, we examined associations of water fluoride and UF levels according to categorical socio-demographic variables using Mann–Whitney U or Kruskal–Wallis tests with post hoc Bonferroni pairwise comparisons. We applied Spearman correlation to examine associations of fluoride variables with continuous demographic variables. We also explored associations of plasma fluoride and UF using Spearman correlation. All analyses applied survey weights, except for these non-parametric tests. Sample weights are not recommended for non-parametric tests as they include rank-based comparisons (e.g., medians, ranks) for which the application of survey weights can lead to biased results [29]. Nevertheless, we explored differences in UF according to sociodemographic variables in the weighted sample for comparison with the unweighted sample.

To examine associations between water fluoride and UF levels, we initially tested covariate-adjusted, linear regression models; however, model assumptions were not satisfied according to regression diagnostics. Therefore, we applied quantile regression, which is more robust to deviations in linear regression assumptions, and more appropriate for non-parametric data as it considers the median rather than mean of the outcome variable [30]. Moreover, quantile regression enables exploration of associations between exposure and outcome variables at different quantiles of the outcome (25th, 50th, and 75th quantiles). Beta coefficients (β) with a 95% confidence interval (CI) were calculated for all models. βs and 95% CIs were rescaled according to an IQR increase in water fluoride levels. We examined separate quantile regression models for children, adolescents, and the overall sample. We also explored whether findings from linear regression with natural log-transformed water fluoride and UF variables differed from quantile regression in supplemental analyses. All models were adjusted for covariates, including age, sex, race/ethnicity, BMI, the ratio of family income to poverty, and urinary creatinine. We included urine creatinine as a separate covariate, rather than as a creatinine-adjusted UF variable. This approach is recommended for multiple regression with urinary biomarker exposure or outcome variables in diverse population-based studies [31]. It allows for associations of an exposure variable and covariates with a urinary chemical biomarker outcome to be parsed independently of any association with urinary creatinine while also simultaneously adjusting for it [31]. We tested interactions of water fluoride by sex, water fluoride by race/ethnicity, and water fluoride by ratio of family income to poverty, to be retained in models if statistically significant. For significant interactions, we computed predictive margins and their 95% CIs in covariate-adjusted quantile regression models. Predictive margins generalize adjusted means to represent the average predicted change across the covariate distribution in the population [32]. These margins also enable the measurement of the absolute difference, rather than relative difference, in the association of an exposure and outcome according to a given variable (i.e., race/ethnicity, sex) [32].

For univariate analyses and quantile regression models that applied survey weights, we utilized survey weights from the mobile exam center visit (i.e., MEC weights). The application of survey weights accounts for the complex NHANES survey design and ensures that results are nationally representative (NCHS) [33]. For analyses that included water fluoride, we reweighted MEC weights using an adjustment factor because we used a variable from a dietary dataset as an exclusion criterion (i.e., tap water drinking habits) [14]. An alpha of 0.05 was considered the threshold for statistical significance. All statistical analyses were performed using STATA version 13.0 and replicated using SAS version 3.81 (Enterprise Edition).

## 3. Results

Participant demographic characteristics are presented in Table 1. The Mean (SD) age was approximately 12.5 (3.9) years, and the distribution of females and males was approximately equal. Most participants identified as Non-Hispanic White (51.84%) and the mean (SD) family income to poverty ratio was 2.52 (1.58). Demographic characteristics in the current study sample were similar to the overall sample of children and adolescents in NHANES 2015–2016 (see Appendix A).

Table 2 presents the distribution of UF and water fluoride levels according to age group. The median (IQR) UF concentration for the overall sample was 0.52 (0.50) mg/L, with higher levels observed among children 0.56 (0.55) mg/L compared to adolescents 0.48 (0.48) mg/L (*p* < 0.001). The median (IQR) household tap water fluoride concentration was 0.39 (0.54) mg/L for participants who consumed tap water. Children had higher levels 0.45 (0.54) mg/L than adolescents 0.35 (0.53) mg/L (*p* = 0.02). Children also had higher UF_CR_ levels (median (IQR) = 0.64 (0.44) mg/L) than adolescents (median (IQR) = 0.56 (0.44) mg/L) (*p* < 0.001) (see Appendix A).

Differences in UF levels based on sociodemographic variables are presented in Table 3. UF levels were higher for males than females, both among children (Median (IQR) = 0.65 (0.57) and 0.48 (0.49), respectively, *p* < 0.001) and adolescents (Median (IQR) = 0.52 (0.47) and 0.43 (0.48), respectively, *p* < 0.001). UF_CR_ levels were also higher among male children (Median (IQR) = 0.67 (0.45) than female children (Median (IQR) = 0.58 (0.44) (*p* = 0.002); however, there were no differences among adolescents, *p* = 0.35).

UF differed according to race/ethnicity among both children (*H* (5) = 37.5, *p* < 0.001) and adolescents (*H* (5) = 42.8, *p* < 0.001) (see Table 3). Specifically, non-Hispanic Black children and adolescents tended to have higher UF levels than all other racial/ethnic groups (*p*s ranged from 0.002 to 0.01), except for Other Race/Multi-Racial (*p* = 0.12 for children and *p* = 0.99 for adolescents). This trend was also apparent in the survey-weighted, nationally representative sample (Appendix A). Interestingly, non-Hispanic Black participants had the greatest proportion whose household tap water fluoride levels ranged from 0.7 to 1.2 mg/L (40.84% of children and 36.59% of adolescents) while for participants from other racial/ethnic backgrounds, at least 70% had water fluoride levels <0.7 mg/L (see Appendix A). There were no differences in UF_CR_ according to race/ethnicity.

UF levels did not differ based on BMI category among children or adolescents (*H* (3) = 1.16, *ρ* = 0.76 and *H*(3) = 0.93, *p* = 0.82, respectively), and UF was not associated with continuous BMI among children (*n* = 1187, *ρ* = 0.01, *p* = 0.62) or adolescents (*n* = 1201, *ρ* = 0.02, *p* = 0.48) either. UF_CR_ was not associated with BMI category; however, UF_CR_ was negatively associated with continuous BMI (*n* = 1187, *ρ* = −0.18, *p* < 0.001 for children; *n* = 1201, *ρ* = −0.09, *p* = 0.003 for adolescents). UF was not associated with ratio of family income to poverty among children (*n* = 1086, *ρ* = −0.03, *p* = 0.29) or adolescents (*n* = 1096, *ρ* = −0.03, *p* = 0.37). Similarly, there were no differences in UF_CR_ according to ratio of family income to poverty.

### 3.1. Associations of Urinary Fluoride with Plasma and Water Fluoride Concentrations

UF was moderately positively correlated with plasma fluoride among children (*n* = 948, *ρ* = 0.58, *p* < 0.001), adolescents (*n* = 1099, *ρ* = 0.51, *p* < 0.001), and the overall sample (*n* = 2083, *ρ* = 0.58, *p* < 0.001). Water fluoride was also positively associated with UF levels for children, adolescents, and the overall sample (Table 4). Notably, the magnitude of association increased from lower to higher quantiles of UF (see Figure 1).

Specifically, for the overall sample, each 1-IQR (0.54 mg/L) increase in water fluoride was associated with a 0.14 mg/L increase in UF at the 25th quantile of UF(β = 0.14, 95% CI; 0.13 to 0.16, *p* < 0.001), a 0.20 mg/L increase in UF at the 50th quantile of UF (β = 0.20, 95% CI; 0.18 to 0.23, *p* < 0.001), and a 0.22 mg/L increase in UF (β = 0.22, 95% CI; 0.18, 0.25, *p* < 0.001) at the 75th quantile of UF. Trends of increasing magnitude of association were consistent across both children and adolescents. Findings from survey-weighted covariate-adjusted linear regression with log-transformed water fluoride and UF were similar (Appendix A).

### 3.2. Associations Between Water Fluoride and UF Concentrations According to Race/Ethnicity

Water fluoride did not significantly interact with sex or income in relation to UF. However, there was an interaction between water fluoride and race/ethnicity in relation to UF such that the magnitude of association was largest for non-Hispanic Black participants (see Figure 2).

For the overall sample of children and adolescents, magnitudes of associations between water fluoride and UF were significantly larger for non-Hispanic Black participants relative to non-Hispanic White participants (i.e., the reference group) at the 50th and 75th quantiles of UF (β = 0.13, 95%CI: 0.03, 0.23, *p* = 0.01 and β = 0.18, 95%CI: 0.05, 0.31, *p* = 0.005, respectively). However, for children, interactions were significant at the 25th quantile (β = 0.13, 95% CI; 0.006, 0.25, *p* = 0.04), the 50th quantile (β = 0.12, 95% CI; 0.006, 0.24, *p* = 0.04), and 75th quantile (β = 0.16, 95% CI; 0.03, 0.30, *p* = 0.02). Interactions were not significant for non-Hispanic Black adolescents (Appendix A). Regarding absolute associations, for non-Hispanic Black participants, each 0.5 mg/L (i.e., approximately 1-IQR) increase in water fluoride was associated with a 0.3 mg/L increase in UF, whereas for Mexican American and non-Hispanic White participants, each 0.5 mg/L increase in water fluoride was associated with 0.19 and 0.18 mg/L increases in UF, respectively. Magnitudes of association were even smaller for other races/ethnicities (Table 5; Figure 2). We observed similar results from survey-weighted covariate-adjusted linear regression examining relative associations between log-transformed water fluoride and UF according to race/ethnicity (Appendix A).

To explore the potential reasons for this interaction, we examined income according to race/ethnicity (see Appendix A) and observed significant differences for both children and adolescents (*H* (5) = 167, *p* < 0.001 and *H* (5) = 148, *p* < 0.001, respectively). Non-Hispanic Black children tended to have the lowest ratio of family income to poverty (median (IQR) = 0.95 (1.33) as compared to all other races/ethnicities. For adolescents, the lowest ratio of family income to poverty was observed for both Mexican American and Non-Hispanic Black participants (Median (IQR) = 1.23 (1.17) and 1.23 (1.4), respectively).

## 4. Discussion

This is the first study to characterize UF levels in a nationally representative sample of children and adolescents residing in the US, while prior studies have focused on plasma and water fluoride levels [25]. Participants were exposed to relatively low levels of fluoride in their tap water; levels that were nearly half of what is recommended for dental caries prevention. Consistently, participants’ UF levels which reflect fluoride intake from all sources were relatively low. UF tended to increase as household tap water fluoride levels increased, with the magnitude of association becoming larger at higher levels of UF. Specifically, each 0.54 mg/L increase in water fluoride was associated with a 0.14 mg/L increase in UF at the 25th quantile of UF (corresponding to 0.31 mg/L of UF), and a 0.20 mg/L increase in UF at the 50th quantile of UF (corresponding to 0.52 mg/L of UF). This suggests that tap water is an important source of fluoride exposure among US youth, particularly at UF levels typical of those living in fluoridated North American communities. Consistently, consumption of tap water (as well as green and black tea) has been shown to predict plasma fluoride levels among youth in NHANES [34].

Studies conducted in Canada have also observed associations between tap water fluoride concentrations and UF. For example, a nationally representative Canadian study of participants aged 3–79 years found that a 1mg/L increase in water fluoride was associated with a 0.48 mg/L increase in specific gravity-adjusted UF (UF_SG_) [35]. A study conducted in the Canadian MIREC cohort also found that each 1 mg/L increase in water fluoride concentration was associated with a 0.44 mg/L increase in UF_SG_ among 2–6-year-old children [36]. However, unlike the current study, the Canadian MIREC study observed a significant interaction of water fluoride by sex, such that the magnitude of association between water fluoride and UF was larger for boys. Taken together, these findings suggest that fluoride may be metabolized differently in girls and boys during early childhood or that they may have different water fluoride consumption patterns.

UF differed according to sociodemographic factors in this study. Most notably, non-Hispanic Black youth tended to have higher UF levels than youth of all other racial/ethnic backgrounds (except for those classified as Other Race/Multi-Racial). Interestingly, non-Hispanic Black youth tended to have lower income than youth of other race/ethnicities. This may have contributed to their higher UF levels, given that community water fluoridation is a targeted intervention among children with low income for reducing income disparities in dental caries [37,38]. However, we cannot confirm that this accounted for the differences in UF according to race/ethnicity observed in this study as we did not have access to data pertaining to participants’ geographic location. Moreover, the magnitude of association between water fluoride and UF was largest among non-Hispanic Black participants. Compared to non-Hispanic White participants, each 0.54 mg/L increase in water fluoride was associated with a 1.3 mg/L greater increase in UF among non-Hispanic Black children and adolescents at the 50th quantile of UF. This suggests that household tap water may be a greater source of fluoride exposure for non-Hispanic Black youth compared to youth from other racial/ethnic backgrounds. Interestingly, approximately 41% of non-Hispanic Black children and 37% of non-Hispanic Black adolescents in the study had household tap water fluoride levels that ranged from 0.7 to 1.2 mg/L, while most other youth had household tap water fluoride levels less than 0.7 mg/L. Consistently, a prior study conducted in NHANES 2013–2014 found that non-Hispanic Black children had the highest proportion of participants with household tap water fluoride levels ranging from 0.7 to 1.2 mg/L; however, non-Hispanic Black adolescents had among the lowest proportion with water fluoride levels in this range [25]. Distributions of water fluoride for non-Hispanic Black youth may differ in that study compared to ours because the recommended water fluoride level was lowered in 2015 to 0.7 mg/L from 0.7 to 1.2 mg/L and the current study includes the 2015–2016 cycle [5]. Nevertheless, these findings suggest that the larger magnitude of association between water fluoride and UF among non-Hispanic Black youth likely reflects exposure to higher community drinking water fluoride levels rather than physiological differences affecting metabolism. Higher fluoride exposure among non-Hispanic Black youth in the US has important public health implications. Numerous studies have shown that Black children suffer significantly higher prevalence and severity of dental fluorosis, an indicator of excess fluoride exposure, compared to their White counterparts [39,40,41]. Interestingly, dental fluorosis is associated with both higher water and plasma fluoride levels among youth in NHANES [42,43], as well as with UF among children exposed to relatively low water fluoride levels in other studies [44]. Racial/ethnic minority youth in the US also face compounded challenges of socioeconomic disparities and systemic oppression in addition to greater exposure to endocrine-disrupting chemicals (EDCs) such as fluoride [45,46,47]. As such, they bare a disproportionate burden of endocrinological and metabolic disorders including diabetes and obesity, as well as female reproductive health disparities [48,49,50,51,52,53,54]. These outcomes have all been associated with fluoride exposure among youth [13,55,56] as well as exposure to other EDCs [57,58,59]. Despite increased fluoride exposure, Black youth in the US, along with Hispanic youth, are still disproportionately affected by dental caries compared to White youth [60,61]. These disparities have been attributed to sociocultural, structural, and familial factors that impact oral healthcare utilization and access to quality care [62].

UF was also associated with other sociodemographic characteristics in this study. Specifically, children tended to have higher UF levels than adolescents and males tended to have higher UF levels than females. Consistently, UF_SG_ was observed to be slightly higher among Canadian children aged 7–11 years than children and adolescents aged 12–18 years living in fluoridated communities [35]. However, UF_SG_ levels were higher among the female compared to male Canadian children and adolescents in that study, although differences were not statistically significant [35]. Nevertheless, a comparison of UF_SG_ among children aged six years or younger in Canada and Mexico found no significant differences according to sex [36]. UF was not significantly associated with BMI or ratio of family income to poverty in this study. Similarly, studies of four-year-old children in Mexico reported no association of UF_SG_ with BMI, and a study of Canadian children aged two to six years reported no association of UFsg with weight [36,63].

This study has several strengths, such as the inclusion of a nationally representative sample and corresponding large sample size which increases generalizability of the findings. Additionally, it includes youth from middle childhood to late adolescence, which enables characterization of fluoride exposure at various stages of development. Furthermore, we adjusted for various sociodemographic variables associated with fluoride exposure/metabolism in our analyses of water fluoride and UF. However, a limitation of this study is that urinary-specific gravity measures were not available during the 2015–2016 cycle of NHANES. Therefore, UF levels were not adjusted for dilution and hydration status may have influenced fluoride concentration measurements. Nevertheless, we adjusted UF for urinary creatinine levels in supplemental analyses and included urinary creatinine as a separate covariate in quantile regression models examining associations of water fluoride with UF. Another limitation is that single-spot UF measurements as opposed to 24-h UF measurements were available for this study which may not capture typical fluoride exposure patterns given that they can be influenced by fluctuations in daily behaviors (i.e., food and beverage consumption). Future nationally representative studies that employ 24-h UF measurement are warranted. Additionally, while tap water provides an important source of fluoride exposure among youth [64], we could not examine associations of fluoride concentrations in other sources (i.e., fluoridated dental products, fluoride from the diet) with UF as these were not assessed in NHANES 2015–2016. However, future studies can examine associations of UF with other potential sources of exposure. Additionally, we did not have access to participants’ geographic locations in this study because geocoded data in NHANES 2015–2016 are restricted access. Therefore, we could not examine whether factors associated with participants’ place of residence may account for differences in fluoride exposure according to race/ethnicity observed in the study. Future studies are needed to explore this question. Lastly, the study included data from the 2015–2016 NHANES cycle, thus capturing patterns of UF exposure at that point in time. Future studies including more recent data are needed to determine whether patterns observed in this study remain consistent into the present time. Future studies examining UF concentrations among US youth in relation to health outcomes are also warranted both for examining child UF as a biomarker and for clarifying potential impacts of childhood and adolescent fluoride exposure from all sources on health outcomes.

## 5. Conclusions

Non-Hispanic Black youth in the US may have greater fluoride exposure than youth of other races/ethnicities, particularly from tap water. Factors contributing to potential racial/ethnic disparities in fluoride exposure within the US warrant further investigation so that they can be mitigated to reduce the potential for harm.

## Figures and Tables

**Figure 1 nutrients-17-00309-f001:**
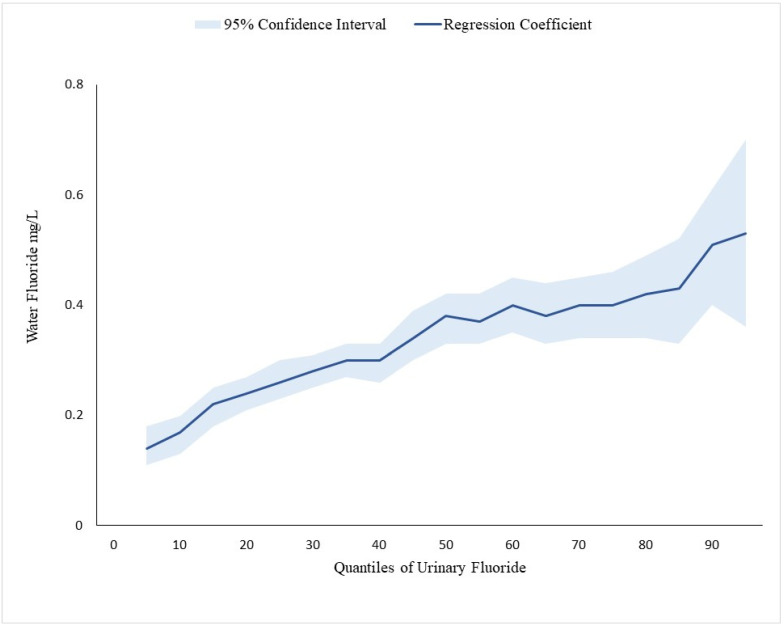
Quantile regression of the association between water fluoride and urinary fluoride. Note: This figure depicts the regression coefficients for associations between water fluoride and urinary fluoride in different quantiles of urinary fluoride.

**Figure 2 nutrients-17-00309-f002:**
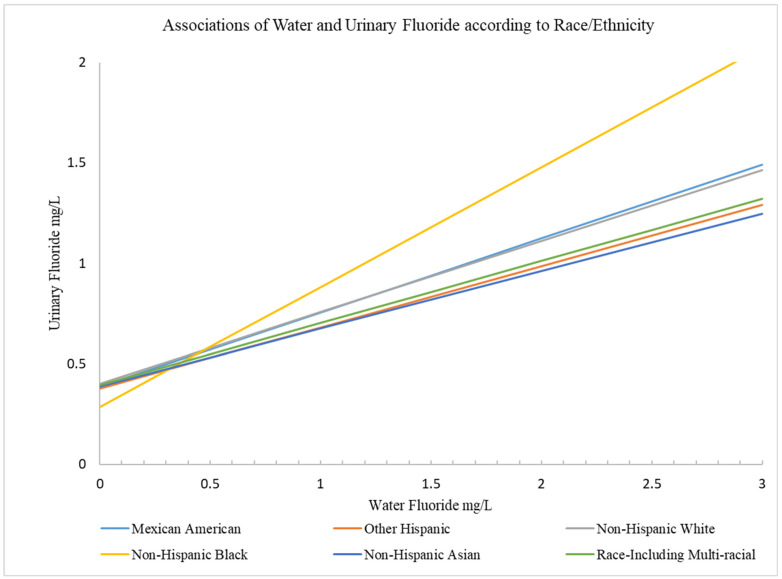
This figure depicts predictive margins representing absolute associations between water fluoride and urine fluoride according to race/ethnicity. The yellow line depicts that the magnitude of association is larger for non-Hispanic Black participants compared to all other race/ethnicity groups.

**Table 1 nutrients-17-00309-t001:** Demographic characteristics of participants included in the final study according to age group.

	Children (6–11 Years)	Adolescent (12–19 Years)	Overall Sample (6–19 Years)
**Total**			
*n* = Unweighted	*n* = 870	*n* = 940	*n* = 1810
N = Weighted	N = 17,903,881	N = 26,672,077	N = 44,575,958
**Age; Mean (SD)**	8.56 (1.71)	15.19 (2.12)	12.53 (3.80)
**Sex; Freq (%) ^a^**			
Male	9,244,919 (51.64)	14,071,602 (52.76)	23,316,522 (52.31)
Female	8,658,961 (48.36)	12,600,475 (47.24)	21,259,437 (47.69)
**Race/Ethnicity; Freq (%) ^a^**			
Mexican American	2,578,723 (14.40)	3,224,979 (12.09)	5,803,702 (13.02)
Other Hispanic	1,471,164 (8.22)	1,987,271 (7.45)	3,458,436 (7.76)
Non-Hispanic White	9,909,625 (55.34)	15,623,437 (58.58)	25,533,062 (57.23)
Non-Hispanic Black	2,042,918 (11.41)	3,280,515 (12.30)	5,323,433 (11.94)
Non-Hispanic Asian	845,027 (4.72)	1,160,240 (4.35)	2,005,268 (4.50)
Other race/multi-racial	1,056,421 (5.90)	1,395,633 (5.23)	2,452,055 (5.50)
**BMI; Mean (SD)**	18.46 (3.93)	24.0 (6.05)	21.78 (5.96)
**Ratio of family income to poverty; Mean (SD)**	2.55 (1.58)	2.61 (1.58)	2.58 (1.58)

Note. Participant demographics for the final study sample included in regression analyses; BMI, Body Mass Index; SD, Standard Deviation; Freq, Frequencies. Weighted study sample (N) calculated by applying NHANES survey MEC weights. ^a^ Reported frequencies are column percentages. All the estimates, mean, SD, frequencies (%) were calculated using NHANES survey MEC weights.

**Table 2 nutrients-17-00309-t002:** Distributions of urine fluoride and water fluoride levels among different age groups.

	Median (IQR)	Mean (SD)	5th, 95th Percentiles
**Children (6–11 years)**			
**Urine Fluoride (mg/L)** *n*(weighted N) = 1191 (22,809,623)	0.56 (0.55)	0.67 (0.54)	0.15,1.55
**Water Fluoride ^a,b^ (mg/L)** *n*(weighted N) = 950 (22,809,624)	0.45 (0.54)	0.48 (0.39)	0.07, 1.00
**Adolescent (12–19 years)**			
**Urine Fluoride (mg/L)** *n*(weighted N) = N = 1217 (32,237,630)	0.48 (0.48)	0.59 (0.44)	0.10, 1.44
**Water Fluoride ^a,b^** *n*(weighted N) = 1047 (32,237,630)	0.35 (0.53)	0.41 (0.33)	0.07, 0.83
**Overall Sample (6–19 years)**			
**Urine Fluoride (mg/L)** *n*(weighted N) = 2408 (55,047,254)	0.52 (0.50)	0.62 (0.49)	0.10, 1.46
**Water Fluoride ^a,b^ (mg/L)** *n*(weighted N) = 1997 (55,047,252)	0.39 (0.54)	0.44 (0.35)	0.07, 0.95

IQR, Inter Quartile Range; SD, Standard Deviation. All estimates, including median, IQR, mean, SD, 5th, 25th, 75th, and 95th percentiles were calculated using NHANES survey MEC weights. ^a^ MEC weights were re-weighted to the dietary sample for analyses including water fluoride. ^b^ Participants who reported that they did not drink the tap water were excluded.

**Table 3 nutrients-17-00309-t003:** Urinary fluoride levels across different sociodemographic factors among youth.

Children (6–11 Years)							Adolescents (12–19 Years)					
Socio-Demographic Factors	N	Mean (SD)	Median (IQR)	Min	Max	*p*-Value	N	Mean (SD)	Median (IQR)	Min	Max	*p*-Value
**Sex**						<0.001						<0.001
Male	595	0.74 (0.51)	0.65 (0.57)	0.10	4.45		628	0.65 (0.48)	0.52 (0.47)	0.10	3.02	
Female	596	0.64 (0.66)	0.48 (0.49)	0.10	10.99		589	0.54 (0.41)	0.43 (0.48)	0.10	3.1	
**Race/Ethnicity**						<0.001						<0.001
Mexican American	278	0.69 (0.78)	0.50 (0.54)	0.10	10.99		266	0.60 (0.50)	0.44 (0.46)	0.10	2.99	
Other Hispanic	168	0.61 (0.38)	0.52 (0.45)	0.10	1.88		148	0.56 (0.42)	0.42 (0.51)	0.10	2.48	
Non-Hispanic White	313	0.64 (0.47)	0.54 (0.54)	0.10	4.45		328	0.57 (0.42)	0.48 (0.44)	0.10	3.1	
Non-Hispanic Black	255	0.85 (0.64)	0.69 (0.68)	0.10	4.48		280	0.69 (0.47)	0.56 (0.51)	0.10	3.02	
Non-Hispanic Asian	98	0.56 (0.46)	0.41 (0.47)	0.10	2.88		122	0.46 (0.39)	0.34 (0.42)	0.10	2.55	
Other race/multi-racial	79	0.68 (0.49)	0.55 (0.58)	0.10	2.61		73	0.64 (0.44)	0.58 (0.50)	0.10	2.52	
**Body Mass Index (BMI)**						0.762						0.817
Underweight	27	0.71 (0.65)	0.48 (0.80)	0.10	2.88		35	0.61 (0.44)	0.50 (0.44)	0.15	1.99	
Normal Weight	720	0.66 (0.48)	0.55 (0.56)	0.10	4.39		658	0.58 (0.43)	0.47 (0.5)	0.10	3.02	
Overweight	194	0.71 (0.60)	0.53 (0.56)	0.10	4.48		231	0.62 (0.48)	0.48 (0.51)	0.10	2.96	
Obese	246	0.74 (0.84)	0.60 (0.54)	0.10	10.99		267	0.61 (0.49)	0.48 (0.45)	0.10	3.1	
Missing	4	0.39 (0.13)	0.39 (0.22)	0.26	0.52		26	0.60 (0.38)	0.46 (0.56)	0.10	1.66	
**Ratio of Family Income to Poverty**	1086	0.68 (0.54)	0.56 (0.54)	0.10	10.99	0.292	1096	0.59 (0.49)	0.48 (0.49)	0.10	3.1	0.373

Note. *n* = 2408; The estimates are unweighted; reported *p*-values were calculated using unweighted non- parametric Mann–Whitney U, Kruskal–Wallis tests, and Spearman correlation.

**Table 4 nutrients-17-00309-t004:** Covariate-adjusted quantile regression of associations between water fluoride and urinary fluoride.

	N	β (95% CI)	*p*-Value
**Children (6–11 years)**			
25th Quantile	870 (17,903,881)	0.14 (0.11,0.17)	<0.001
50th Quantile	870 (17,903,881)	0.18 (0.16, 0.21)	<0.001
75th Quantile	870 (17,903,881)	0.19 (0.15, 0.24)	<0.001
**Adolescent (12–19 years)**			
25th Quantile	940 (26,672,077)	0.14 (0.10, 0.15)	<0.001
50th Quantile	940 (26,672,077)	0.19 (0.12, 0.22)	<0.001
75th Quantile	940 (26,672,077)	0.25 (0.19, 0.30)	<0.001
**Overall (6–19 years)**			
25th Quantile	1810 (44,575,958)	0.14 (0.13, 0.16)	<0.001
50th Quantile	1810 (44,575,958)	0.20 (0.18, 0.23)	<0.001
75th Quantile	1810 (44,575,958)	0.22 (0.18, 0.25)	<0.001

Participants who reported that they did not drink the tap water were excluded; β Coefficients and 95% CIs are rescaled according to an IQR (i.e., 0.54 mg/L for children; 0.53 mg/L for adolescents; 0.54 for overall) increase in water fluoride levels. The β estimates, 95% CIs and *p*-values were calculated using NHANES survey MEC weights. MEC weights were re-weighted to the dietary sample for regression analyses. All models are adjusted for age, sex, race/ethnicity, BMI, ratio of family income to poverty, and urine creatinine levels; unweighted samples sizes are *n* = 870 for children, *n* = 940 for adolescents, *n* = 1810 for the overall sample.

**Table 5 nutrients-17-00309-t005:** Predictive margins for urinary fluoride (mg/L) according to race/ethnicity at different levels of water fluoride (mg/L).

Water Fluoride (mg/L)	Mexican American Predictive Margins (95% CI)	Other Hispanic Predictive Margins (95% CI)	Non-Hispanic White Predictive Margins (95% CI)	Non-Hispanic Black Predictive Margins (95% CI)	Non-Hispanic Asian Predictive Margins (95% CI)	Mixed RacIncluding Multi-Racial Predictive Margins (95% CI)
0	0.39 (0.32, 0.45)	0.38 (0.28, 0.48)	0.40 (0.37, 0.44)	0.29 (0.18, 0.39)	0.39 (0.23, 0.55)	0.39 (0.27, 0.51)
0.5	0.57 (0.52, 0.62)	0.53 (0.47, 0.59)	0.58 (0.56, 0.60)	0.58 (0.53, 0.63)	0.53 (0.45, 0.61)	0.55 (0.47, 0.63)
1	0.76 (0.70, 0.81)	0.68 (0.57, 0.80)	0.76 (0.71, 0.81)	0.88 (0.79, 0.98)	0.68 (0.47, 0.88)	0.70 (0.53, 0.88)
1.5	0.94 (0.85, 1.03)	0.83 (0.63, 1.03)	0.93 (0.85, 1.02)	1.18 (1.00, 1.35)	0.82 (0.46, 1.17)	0.86 (0.56, 1.15)
2	1.12 (1.00, 1.25)	0.99 (0.70, 1.27)	1.11 (0.99, 1.23)	1.48 (1.22, 1.74)	0.96 (0.45, 1.48)	1.01 (0.59, 1.44)
2.5	1.31 (1.14, 1.47)	1.14 (0.77, 1.51)	1.29 (1.14, 1.44)	1.78 (1.43, 2.12)	1.10 (0.43, 1.78)	1.17 (0.61, 1.72)
3	1.49 (1.29, 1.70)	1.29 (0.83, 1.75)	1.46 (1.28, 1.65)	2.07 (1.64, 2.51)	1.23 (0.41, 2.08)	1.32 (0.64, 2.00)

N = 44,575,958 (Unweighted *n* = 1810) and includes the overall sample of children and adolescents; Predictive margins were computed from survey-weighted quantile regression models adjusted for age, sex, race/ethnicity, BMI, ratio of family income to poverty, and urine creatinine levels.

## Data Availability

Data from National Health and Nutrition Examination Surveys (NHANES) are publicly available. The raw data supporting the conclusions of this article will be made available by the authors on request.

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
