# Peer review of "Urinary Fluoride Levels Among Youth in the National Health and Nutrition Examination Survey (NHANES) 2015–2016: Potential Differences According to Race"

_nutrients, 2025, doi:10.3390/nu17020309_

Round 1
Reviewer 1 Report
Comments and Suggestions for Authors
1. What was the scientific problem? The connection between urinary fluoride (UF) and concentrations in drinking water and plasma? This has been known for decades, as the body mainly excretes fluoride through urine. The hypothesis that plasma and water fluoride levels would each be positively associated with UF is not revealing. However, the connection between UF and sociodemographic parameters has not been justified and appropriately discussed. There are some statements, but there is no at least hypothetical explanation for them. What could be the relationship between family income to poverty, and fluoride excretion in urine?
2. Why is there such a large difference in UF between the groups when the differences in plasma are negligible?
3. There is no data on other sources of fluoride in the diet.
4. The authors only analyzed fluoride concentration in household water. Is water from the participants’ place of residence (homes) the only water consumed by the subjects? How can we be sure of this?
5. Lines 299-301: „Taken together, these findings suggest that fluoride may be metabolized differently in girls and boys during early childhood, or that they may have different tap water consumption patterns”. This is debatable because we do not know the fluoride intake from other sources.
6. Lines 304-307: „Interestingly, non-Hispanic Black youth tended to have lower income than youth of other races/ethnicities. This may have contributed to their higher UF levels, given that community fluoridation is a targeted intervention for reducing income disparities in dental caries”. Does this mean that water fluoridation is more intense in non-Hispanic Black populations than in others? How does it work technically since everyone drinks water from the same source? Are there water sources in the USA designated only for certain ethnic groups? How is it verified that only non-Hispanic Black people drink water from a particular source and not others?
7. Differences without statistical significance cannot be used as the basis for concluding that sex or age influence fluoride excretion.
8. Among sociodemographic factors, only race mattered. However, non-Hispanic Black individuals consumed water with, on average, higher fluoride concentrations than others, which explains not the impact of race, but rather the relationship between the consumed dose and excretion. Therefore, the conclusion is clear: since there is a correlation between fluoride concentration in drinking water and its excretion in urine, and non-Hispanic Black individuals drink water with higher fluoride concentrations, it is no surprise that they have higher UF compared to others. Therefore, there is no racial impact, but the effect of water fluoride concentration rather, which we have known for decades.
9. In the Limitations section, the authors do not mention the main limitation, namely fluoride intake from other sources and potential exposure to industrial emissions. From a scientific point of view, the paper is not groundbreaking. If the scientific problem was to investigate the correlations only, then it is hard to speak of novelty.
10. The authors limited themselves to summarizing old data that does not bring anything we didn't already know. The influence of race would be very interesting, but it was not demonstrated. They did not even attempt to explain what potential biochemical mechanisms might cause non-Hispanic Black individuals to excrete more fluoride than others.
Author Response
Reviewer 1
- What was the scientific problem? The connection between urinary fluoride (UF) and concentrations in drinking water and plasma? This has been known for decades, as the body mainly excretes fluoride through urine. The hypothesis that plasma and water fluoride levels would each be positively associated with UF is not revealing. However, the connection between UF and sociodemographic parameters has not been justified and appropriately discussed. There are some statements, but there is no at least hypothetical explanation for them. What could be the relationship between family income to poverty, and fluoride excretion in urine?
Thank you for this comment. The scientific problem is that patterns of urinary fluoride concentrations have never been characterized in a nationally representative sample of United States children. This has been highlighted as an important research need in the recently published meta-analysis from the National Toxicology Program (NTP). The NTPs meta-analysis examined early life fluoride exposure and child IQ; they emphasized the importance of having a nationally representative sample of urinary fluoride data to help contextualize their recent findings. The report states “no nationally representative urinary fluoride levels are available, hindering application of these findings to the US population.” We have now included this in the introduction to clarify the research problem as well as appropriately justify the connection between UF and sociodemographic parameters on lines 54 – 60 in the track change version.
Fluoride Exposure and Children’s IQ Scores: A Systematic Review and Meta-Analysis | Pediatrics | JAMA Pediatrics | JAMA Network
We did not have specific hypotheses as to what associations may be observed between UF and sociodemographic factors as these analyses were exploratory. We now clarify this on lines 65-66.
- Why is there such a large difference in UF between the groups when the differences in plasma are negligible?
We are not entirely sure what group differences you are referring to; however, Table S3 is the only table in which we compare differences in plasma and urine fluoride according to subgroup. The differences in plasma fluoride between children and adolescents may be smaller than for UF because UF better reflects fluoride excretion which can be influenced by individual differences in fluoride metabolism as well as hydration status (which can be influenced by sociodemographic factors) more so than plasma fluoride which better reflects fluoride absorption. (see Buzalaf, M.A., Whitford, G.M., 2011. Fluoride metabolism. Monogr. Oral Sci. 22, 20–36).
- There is no data on other sources of fluoride in the diet.
You are correct in that we did not include specific sources of fluoride in the diet as NHANES did not have this data available. That said, urinary fluoride captures fluoride intake from all sources. We now clarify this on line 52.
- The authors only analyzed fluoride concentration in household water. Is water from the participants’ place of residence (homes) the only water consumed by the subjects? How can we be sure of this?
Thank you for this question. NHANES only provided water fluoride measurements that were taken directly from participant’s taps in their homes. We have added some text regarding this on lines 133-135. You are correct that we cannot be sure that this was the only water consumed by participants that contains fluoride. In fact, all food and beverages that are reconstituted with the fluoridated tap water will also contain fluoride. As mentioned above, urinary fluoride concentration measurements will capture fluoride intake from all sources.
- Lines 299-301: „Taken together, these findings suggest that fluoride may be metabolized differently in girls and boys during early childhood, or that they may have different tap water consumption patterns”. This is debatable because we do not know the fluoride intake from other sources.
Thank you for this comment. That is a good point. However, in this paragraph we are referring to findings examining associations of water and urinary fluoride specifically. That said, we have modified this statement slightly to clarify that findings may point to differences in water fluoride consumption patterns in general rather than specifically from tap water, as the studies we refer to did not necessarily examine tap water fluoride directly; some used municipal water treatment data (line 335).
- Lines 304-307: „Interestingly, non-Hispanic Black youth tended to have lower income than youth of other races/ethnicities. This may have contributed to their higher UF levels, given that community fluoridation is a targeted intervention for reducing income disparities in dental caries”. Does this mean that water fluoridation is more intense in non-Hispanic Black populations than in others? How does it work technically since everyone drinks water from the same source? Are there water sources in the USA designated only for certain ethnic groups? How is it verified that only non-Hispanic Black people drink water from a particular source and not others?
Thank you for this question. Communities with low income tend to be targeted with community fluoridation to reduce disparities in dental caries related to income. Therefore, if more non-Hispanic Black participants tend to live in lower income communities, they may receive more community water fluoridation. However, given that we did not have access to geocoded data in NHANES because it is restricted from public access, we can not verify this directly. We have now added additional clarification and context regarding this on lines 342-344.
- Differences without statistical significance cannot be used as the basis for concluding that sex or age influence fluoride excretion.
We agree; however, in our study we did find statistically significant differences in urinary fluoride according to age and sex. In our results we report:
UF levels were higher for males than females, both among children (Median (IQR) = 0.65 (0.57) and 0.48 (0.49) respectively, p <0.001) and adolescents (Median (IQR) = 0.52 (0.47) and 0.43 (0.48) respectively, p <0.001).
The median (IQR) UF concentration for the overall sample was 0.52 (0.50) mg/L, with higher levels observed among children 0.56 (0.55) mg/L compared to adolescents 0.48 (0.48) mg/L ( p <0.001).
- Among sociodemographic factors, only race mattered. However, non-Hispanic Black individuals consumed water with, on average, higher fluoride concentrations than others, which explains not the impact of race, but rather the relationship between the consumed dose and excretion. Therefore, the conclusion is clear: since there is a correlation between fluoride concentration in drinking water and its excretion in urine, and non-Hispanic Black individuals drink water with higher fluoride concentrations, it is no surprise that they have higher UF compared to others. Therefore, there is no racial impact, but the effect of water fluoride concentration rather, which we have known for decades.
Thank you for this comment. We agree that race is unlikely causing higher urinary fluoride levels among non-Hispanic Black participants, and that their urinary fluoride levels are likely higher because their exposure to fluoride in drinking water is higher. We’ve provided additional clarity regarding this on line 362 and have modified our conclusion statement for clarity as well.
- In the Limitations section, the authors do not mention the main limitation, namely fluoride intake from other sources and potential exposure to industrial emissions. From a scientific point of view, the paper is not groundbreaking. If the scientific problem was to investigate the correlations only, then it is hard to speak of novelty.
We agree that the paper is not groundbreaking; however, it does provide essential information that can help to inform risk assessment of child fluoride exposure in the United States, as well as address a pertinent public health need highlighted by the National Toxicology Program.
We had modified our limitations section on lines 407- 410 to state that we did not examine associations of fluoride from other sources with UF; although, we note that water fluoride is a primary source of fluoride exposure among youth. Exposure from industrial emissions would be rare among children and adolescents residing in the United States.
(see Fluoride: exposure and relative source contribution analysis. US Environmental Protection Agency. 2010)
- The authors limited themselves to summarizing old data that does not bring anything we didn't already know. The influence of race would be very interesting, but it was not demonstrated. They did not even attempt to explain what potential biochemical mechanisms might cause non-Hispanic Black individuals to excrete more fluoride than others.
While we appreciate your perspective and understand that data from NHANES 2015-2016 is not new, it only became publicly available in November 2022, and it is the only cycle that published urinary fluoride levels among youth. Moreover, patterns of urinary fluoride with sociodemographic variables have never been published for a nationally representative sample residing in the United States. We now add a limitation regarding having used an earlier cycle of NHANES data on lines 415-418. Regarding the influence of race, the cross-sectional nature of the NHANES study design precludes examination of a casual influence of race on UF levels. Therefore, we could not examine whether race influenced urinary fluoride levels in any survey year. To our knowledge, there are not biochemical mechanisms pertaining to being Black that would influence fluoride metabolism and excretion. We emphasized this on lines 360-363 when we state:
..findings suggest that the larger magnitude of association between water fluoride and UF among non-Hispanic Black youth likely reflects exposure to higher community drinking water fluoride levels rather than physiological differences affecting metabolism.
Reviewer 2 Report
Comments and Suggestions for Authors
Dear Authors,
Thank you very much for this interesting study and manuscript. Some improvements might help to increase the quality and interest.
Title: the term NHANES is not a very common abbreviation. Please clarify and change the title. I would use National Health and Nutrition Examination Survey also in the title.
Abstract: Please explain the term NHANES also in the abstract. Please add at least at the beginning the full name (National Health and Nutrition Examination Survey). Thank you. Please give a clinical conclusion at the end of the abstract. This might increase reader´s interest in your work.
Introduction: Please state a clear hypothesis and null hypothesis. Thank you.
Material and Methods:
Please give more information about the included patients (region, city, state),
Did you record oral hygiene procedures and possible fluoride intact? Please clarify. Please give more information about the ethical approval and the written consent of the included patients. Thank you.
Did you ask for additional fluoride intake (milk, salt, and so on)? Please add.
Please give more information about measuring the fluoride concentrations in urine, water and plasma. This might help understand the complete study protocol.
Regional differences and distribution should be included.
Discussion: Please discuss the influence of other fluoride sources as well.
Statements at the end: Please include the Institutional Review Board Statement, the Informed Consent Statement, the Data Availability Statement and the Conflicts of Interest. Please add.
Author Response
Dear Authors,
Thank you very much for this interesting study and manuscript. Some improvements might help to increase the quality and interest.
Title: the term NHANES is not a very common abbreviation. Please clarify and change the title. I would use National Health and Nutrition Examination Survey also in the title.
Thank you for this comment. We have changed this in the title as well as the abstract.
Abstract: Please explain the term NHANES also in the abstract. Please add at least at the beginning the full name (National Health and Nutrition Examination Survey). Thank you. Please give a clinical conclusion at the end of the abstract. This might increase reader´s interest in your work.
We now explain the term NHANES in the abstract. We agree that including a clinical conclusion will be important and have now added this at the end of the abstract.
Introduction: Please state a clear hypothesis and null hypothesis. Thank you.
We have modified our hypothesis statement on lines 65-67 accordingly. However, please note that we did not have specific hypotheses pertaining to associations of UF with sociodemographic variables as these analyses were exploratory. We now clarify this in our revised hypothesis statement.
Material and Methods:
Please give more information about the included patients (region, city, state)
We have added more details about the participants included in the National Health and Nutrition Examination Survey on lines 71-81; however, we do not have information pertaining to the specific geographic location of participants as that is restricted from public access.
Did you record oral hygiene procedures and possible fluoride intact? Please clarify. Please give more information about the ethical approval and the written consent of the included patients. Thank you.
Thank you for this question. Unfortunately, NHANES did not assess fluoride intake from fluoridated dental products. We now include this as a limitation on lines 407-410.
We now include a statement about ethical approval in NHANES on lines 82-83, as well as a statement regarding informed consent on lines 76-81.
Did you ask for additional fluoride intake (milk, salt, and so on)? Please add.
Unfortunately, NHANES did not assess other sources of fluoride intake aside from household tap water; although urinary fluoride captures all sources of fluoride intake. Please note that the United States does not add fluoride to milk or salt. We now provide a reference for sources of fluoride exposure in the US on line 408.
Please give more information about measuring the fluoride concentrations in urine, water and plasma. This might help understand the complete study protocol.
Thank you for this comment. We have now added more details about fluoride measurement protocols for urine, water, and plasma to these sections (lines 101-135).
Regional differences and distribution should be included.
Thank you for this comment. We agree it would be ideal to have this information; however, geocoded data that includes region the study participant is located in is restricted from public access in NHANES. We note this in the discussion on lines 410-414.
Discussion: Please discuss the influence of other fluoride sources as well.
NHANES unfortunately did not have data on other sources of fluoride; although urinary fluoride levels reflect total fluoride intake. We now emphasize this in the discussion on lines 315-316. Additionally, we’ve now added a limitation to the discussion regarding this on lines 407-410.
Statements at the end: Please include the Institutional Review Board Statement, the Informed Consent Statement, the Data Availability Statement and the Conflicts of Interest. Please add.
We have included our IRB statement on lines 96-97; the informed consent statement is on lines 76-81; the data availability statement is on lines 442-444; the conflicts of interest statement is on lines 445.
Round 2
Reviewer 2 Report
Comments and Suggestions for Authors
Dear authors,
thank you for adressing the raised points.